# F-Box Proteins and Cancer

**DOI:** 10.3390/cancers12051249

**Published:** 2020-05-15

**Authors:** Kanae Yumimoto, Yuhei Yamauchi, Keiichi I. Nakayama

**Affiliations:** Department of Molecular and Cellular Biology, Medical Institute of Bioregulation, Kyushu University, 3-1-1 Maidashi, Higashi-ku, Fukuoka, Fukuoka 812-8582, Japan; tnkana@bioreg.kyushu-u.ac.jp (K.Y.); yamauchi_yuhei3x@bioreg.kyushu-u.ac.jp (Y.Y.)

**Keywords:** ubiquitin ligase, F-box protein, mutation, cancer

## Abstract

Controlled protein degradation is essential for the operation of a variety of cellular processes including cell division, growth, and differentiation. Identification of the relations between ubiquitin ligases and their substrates is key to understanding the molecular basis of cancer development and to the discovery of novel targets for cancer therapeutics. F-box proteins function as the substrate recognition subunits of S-phase kinase-associated protein 1 (SKP1)−Cullin1 (CUL1)−F-box protein (SCF) ubiquitin ligase complexes. Here, we summarize the roles of specific F-box proteins that have been shown to function as tumor promoters or suppressors. We also highlight proto-oncoproteins that are targeted for ubiquitylation by multiple F-box proteins, and discuss how these F-box proteins are deployed to regulate their cognate substrates in various situations.

## 1. Introduction

Ubiquitylation-dependent proteasomal degradation of target proteins is an irreversible reaction that contributes to the regulation of many eukaryotic cellular processes—such as cell division, growth, and differentiation—at the level of selective protein turnover [1]. Impaired function of the ubiquitin-proteasome system often results in the aberrant stabilization of proto-oncogenic protein substrates that can eventually lead to carcinogenesis [2]. Conversely, abnormal up-regulation of the expression of certain ubiquitin ligases can result in the excessive degradation of tumor suppressor proteins and thereby also give rise to carcinogenesis. It is therefore important that the abundance of ubiquitin ligases that target promoters or suppressors of tumorigenesis be strictly controlled.

Ubiquitylation occurs via a sequence of enzymatic events in which the small protein ubiquitin is activated by linkage to an E1 (ubiquitin-activating) enzyme and is then transferred first to an E2 (ubiquitin-conjugating) enzyme and then to a free amine group either at the NH_2_-terminus or on an internal lysine residue of a substrate protein, with the transfer to the substrate being mediated by an E3 (ubiquitin ligase) enzyme. HECT (homologous to E6AP COOH-terminus)-type E3 enzymes catalyze ubiquitylation by first forming an E3-ubiquitin thioester intermediate, whereas RING (really interesting new gene) finger-, U box-, and plant homeodomain (PHD)-type E3s do not appear to form such an intermediate. In mammals, the largest family of E3 ubiquitin ligases consists of Cullin-RING ligases, of which SKP1–CUL1–F-box protein (SCF) complexes are the best characterized. Each SCF complex comprises SKP1 (S-phase kinase-associated protein 1), CUL1 (Cullin 1), RBX1 (RING-box protein 1; also known as ROC1, or regulator of Cullin 1), and one of a variety of F-box proteins [3]. F-box proteins are responsible for substrate recognition by each SCF complex, and they harbor two key functional domains: The F-box domain, which mediates association with the other components of the complex via direct interaction with SKP1, and the COOH-terminal domain, which interacts with substrates [4]. On the basis of their different COOH-terminal regions, F-box proteins are categorized into three classes: FBXW (containing a WD40-repeat domain), FBXL (containing a leucine-rich-repeat domain), and FBXO (containing another type of protein interaction domain or no recognizable domain) (Figure 1) [4]. Overexpression or mutations (deletions or point mutations) of certain F-box proteins have been associated with cancer progression [5]. In addition, mutations of the amino acid sequences of substrate proteins that are recognized by ubiquitin ligases, as well as perturbations that affect the posttranslational modification of such “degrons,” may contribute to cancer development [6].

In this review, we summarize the oncogenic and oncosuppressive roles of specific F-box proteins. We also highlight proto-oncoproteins that are targeted for ubiquitylation by multiple F-box proteins as well as discuss how such F-box proteins are deployed to regulate such substrates in a manner dependent on the localization or function of the latter.

## 2. Representative F-Box Proteins Related to Cancer

### 2.1. FBXO1 (Cyclin F)

Cyclin F (FBXO1) was first identified in 1994 as the founding member of the F-box protein family [7]. It also belongs to the cyclin family of proteins and contributes to regulation of various cell cycle-related processes including centrosome duplication [8], maintenance of genome stability [9], and DNA replication and repair [10]. The amounts of cyclin F mRNA and protein oscillate during the cell cycle [7], with the amount of the mRNA increasing in S phase, peaking in G_2_ phase, and declining again in M phase. In contrast to other cyclins that cooperate with cyclin-dependent kinases (CDKs) to phosphorylate protein substrates and thereby control cell cycle events, cyclin F directly interacts with and mediates the ubiquitylation of its substrates in a manner independent of CDK activity [8,10,11].

SCF^Cyclin F^ mediates the degradation of centrosomal protein of 110 kDa (CP110), a promoter of centrosome duplication, a process that must occur only once during each cell cycle [8]. The timely degradation of CP110 controlled by SCF^Cyclin F^ is therefore key to prevention of centrosome duplication more than once in the cell cycle. SCF^Cyclin F^ also targets cell division cycle 6 (CDC6), a protein that contributes to DNA replication, for ubiquitylation and subsequent degradation [9]. Such degradation of CDC6 prevents DNA re-replication or overreplication, which is highly disruptive to DNA integrity and increases the risk of genome instability. Another target of SCF^Cyclin F^ is stem-loop binding protein (SLBP), which contributes to the processing, translation, and degradation of mRNAs for canonical histones. The abundance of SLBP closely matches the demand for core histone proteins, and it declines rapidly during the G_2_ phase of the cell cycle as a result of SCF^Cyclin F^-dependent degradation [12]. SCF^Cyclin F^ controls assembly of the mitotic spindle by mediating degradation of nucleolar and spindle-associated protein 1 (NuSAP1), a protein that contributes to the organization of microtubules. In addition, degradation of the three activator E2F transcription factors (E2F1, E2F2, and E2F3A) in late S and G_2_ phases was recently shown to be mediated by SCF^Cyclin F^ [13,14]. These various findings suggest that SCF^Cyclin F^ ensures accurate cell cycle progression in the G_2_ phase by targeting for degradation proteins that function in S phase.

Cyclin F also plays an important role in the cellular response to DNA damage. Cells depleted of cyclin F initiate checkpoint signaling after exposure to radiation, but they fail to maintain G_2_ arrest and progress prematurely into mitosis [11]. Moreover, SCF^Cyclin F^ mediates the ubiquitylation of RRM2 [10], a subunit of ribonucleotide reductase, which catalyzes the conversion of ribonucleotides to deoxyribonucleotides. In response to the genotoxic induction of DNA damage, cyclin F is destabilized in a manner dependent on the protein kinase ATR, leading to RRM2 accumulation and deoxyribonucleotide production to facilitate DNA repair. In addition, cyclin F interacts with the transcription factor b-Myb via its cyclin-box domain and thereby suppresses cyclin A-mediated activation of b-Myb [11] in response to DNA damage signaling. Finally, SCF^Cyclin F^ ubiquitylates exonuclease 1 and is thereby thought to prevent unnecessary DNA resection under stress conditions [15].

Given the characteristics of the substrates of SCF^Cyclin F^, cyclin F might be expected to serve as a tumor suppressor. Indeed, overexpression of cyclin F in cultured cells was found to induce G_2_ arrest and to prevent initiation of mitosis [7]. It has also been shown to limit centrosome duplication, with centrosome amplification being a hallmark of cancer. Furthermore, under conditions of metabolic stress, SCF^Cyclin F^ was found to be induced and to mediate polyubiquitylation of RBP-Jκ, a mediator of Notch signaling, resulting in attenuation of the RBP-Jκ-dependent oncogenic function of the R132H mutant of isocitrate dehydrogenase 1 (IDH1) [16]. Consistent with its expected role in tumor suppression, the expression of cyclin F has been found to be down-regulated in hepatocellular carcinoma, with the extent of this down-regulation correlating with a range of prognostic markers and having been proposed as a prognostic indicator in this disease [17].

### 2.2. FBXL1 (SKP2)

SKP2 (also known as FBXL1) was first identified as a protein associated with cyclin A [18]. It was subsequently shown to mediate the polyubiquitylation and consequent degradation of CDK inhibitors such as p27*^KIP1^* [19,20,21,22,23], p57*^KIP2^* [24], and p21*^CIP1^* [25,26], and thereby to regulate the G_1_-S and G_2_-M transitions of the cell cycle [22,27]. Although mice deficient in SKP2 are viable, cells of the mutant animals contain markedly enlarged nuclei with polyploidy and multiple centrosomes as well as manifest a reduced growth rate and increased level of apoptosis [21]. These phenotypes are reversed by additional ablation of p27*^KIP1^*, suggesting that this CDK inhibitor is the major substrate targeted by SCF^SKP2^ [22]. Several types of mutation and amplification of *SKP2* have been detected in various types of cancer, including cervical, endometrial, adrenocortical, ovarian, breast, and non-small-cell lung cancer [28]. Furthermore, the expression level of SKP2 is negatively associated with the survival of cancer patients—including those with prostate cancer [29], breast cancer [30,31], or lung cancer [32]—suggesting that *SKP2* acts as an oncogene. Consistent with this latter notion, overexpression of SKP2 in mice promotes the development of prostate gland dysplasia [33] and T cell lymphoma [34], whereas ablation of *Skp2* impairs these processes [35,36]. Other cell cycle regulators—including p130 [37], Forkhead box protein O1 (FOXO1) [38], cyclin E [21], CDK9 [39], Recombination activating gene 2 (RAG2) [40], and b-Myb [41]—have also been identified as degradative substrates for SCF^SKP2^.

In addition to the proteolytic function of SKP2, recent studies have revealed that SKP2 exerts its oncogenic activity by mediating the conjugation of lysine-63 (K63)-linked polyubiquitin chains to target proteins such as protein kinase B (PKB; also known as AKT) [42], Yes-associated protein (YAP) [43], Twist transcriptional factor [44], Nijmegen breakage syndrome 1 (NBS1) [45], Ras related GTP binding A (RAG-A) [46], Breakpoint cluster region-Abelson (BCR-ABL) [47], human MutT homolog 1 (MTH1) [48], and Aurora-B [49], which results in the stabilization or alters the localization of these proteins or promotes the recruitment of interactors. The K63-linked ubiquitylation of the protein kinase AKT by SCF^SKP2^ is stimulated by growth factors and cytokines, and it facilitates the recruitment of AKT to the plasma membrane, with its resulting hyperactivation leading to up-regulation of glycolysis and promotion of tumorigenesis [42,50]. A recent study also indicates that SCF^SKP2^-mediated AKT ubiquitylation contributes to chemoresistance of cancer cells [51]. The phosphatidylinositol 3-kinase (PI3K) inhibitor Buparlisib (BKM-120) was thus shown to promote SCF^SKP2^-mediated ubiquitylation and reactivation of AKT in a subset of triple-negative breast cancer cell lines, and depletion of SKP2 attenuated the growth of such cells that were resistant to BKM-120. K63-linked ubiquitylation of the transcription factor YAP by SCF^SKP2^ promotes its nuclear localization in a manner independent of Hippo signaling and thereby enhances its transcriptional activity and growth-promoting function [43]. YAP, in turn, promotes the p300-mediated acetylation and cytoplasmic retention of SKP2, resulting in excessive degradation of the pro-apoptotic factors FOXO1 and FOXO3, the development of polyploidy and genomic instability, and oncogenesis [52]. These findings raise the possibility that SKP2 and YAP each reciprocally enhance the oncogenic function of the other.

Given the roles of SKP2 in carcinogenesis and cancer progression, inactivation of SKP2 is a potentially promising approach to cancer treatment. Several inhibitors of SKP2 have been developed [53]. Compound 25 (also known as SZL-P1–41), for example, disrupts the interaction between SKP2 and SKP1 and thereby promotes disassembly of the SCF^SKP2^ complex [54], whereas compound A (also known as CdpA) disrupts the interaction between SKP2 and p27*^KIP1^* [55]. Some agents target the interaction between SKP2 and the protein CDK subunit 1 (CKS1), which is essential for the recognition of p27*^KIP1^* by SKP2 [56,57,58]. A compound known as Dioscin was recently shown to attenuate SKP2 expression [59], in part by enhancing the interaction between SKP2 and Fizzy-related, Hct1 homolog (CDH1), a cofactor of APC/C (anaphase-promoting complex/cyclosome) that mediates the ubiquitylation and degradation of SKP2 [60,61]. Further investigation is required to determine whether these various inhibitors are effective in cancer treatment.

### 2.3. FBXW7

F-box/WD repeat-containing protein 7 (FBXW7; also known as Fbw7, SEL-10, hCDC4, or hAgo) was first discovered as a negative regulator of the LIN-12 (Notch) signaling pathway in *Caenorhabditis elegans* by genetic screening [62]. It was later shown to negatively regulate cell cycle progression, stem cell maintenance, and differentiation of multiple cell types by mediating the degradation of various substrates in a phosphorylation-dependent manner [63,64,65]. FBXW7 recognizes a conserved phosphorylation motif known as the Cdc4 phosphodegron (CPD), which typically consists of φXφφφ(T/S)PPX(T/S/D/E) (where φ represents a hydrophobic amino acid) [66]. About 90 proteins have been identified to date as substrates of FBXW7 [67], many of which are proto-oncoproteins including a transcription factor c-Myc [68,69], a transmembrane receptor Notch [62,70,71], a transcription factor c-Jun [72,73], cyclin E [74,75,76], Krüppel-like factor 5 (KLF5) [77,78], and Myeloid cell leukemia 1 (MCL1) [79,80]. 

Among the ~70 F-box protein genes identified in humans, *FBXW7* has the highest mutation frequency in cancer [67], indicative of the importance of the encoded protein as a tumor suppressor in cancer development. Cancer-associated *FBXW7* mutations are most frequent in T cell acute lymphoblastic leukemia (T-ALL), followed by precursor T cell lymphoblastic lymphoma (T-LBL), endometrial carcinoma, small intestine carcinoma, and large intestine carcinoma. Such mutations of *FBXW7* are concentrated in the codons for three arginine residues (Arg^465^, Arg^479^, and Arg^505^) in the WD40-repeat domain responsible for substrate binding. These arginine residues form a positively charged pocket at the inner rim of the β-propeller channel of the protein, and they form charge-stabilized hydrogen bonds with the substrate by enveloping the negatively charged phosphate group at the “0” position of the CPD [66,81].

In addition to loss-of-function mutations in the substrate binding domain of FBXW7, mutations of FBXW7 substrates have been found to be associated with cancer and to concentrate in the CPD. Activating mutations of *NOTCH1* have thus been identified in ~50% to 60% of T-ALL [82] and precursor T-LBL [83,84] cases, and ~10% of these mutations affect the codon for Pro^2514^ in the CPD of the encoded protein and are predicted to abrogate the interaction with FBXW7. In addition, in ~43% of individuals with diffuse large B cell lymphoma (DLBCL) positive for nonsense or missense mutations of *MYC*, the mutations are located in the CPD sequence [67], suggesting that impaired degradation of c-Myc by FBXW7 is a key mechanism underlying c-Myc accumulation in DLBCL. KLF5, another substrate of FBXW7, is necessary for the initial steps of tumor development in *Apc*^Min^ and *Apc*^Min^*KRAS*^G12V^ mouse models of colorectal cancer [85,86], and a large proportion of the mutations of KLF5 in cancer patients are concentrated in its CPD [67].

Certain small compounds have been found to activate SCF^FBXW7^ directly or indirectly. Oridonin increases the expression of FBXW7 and also activates glycogen synthase kinase–3 (GSK-3), which mediates the phosphorylation of c-Myc and thereby promotes its turnover, in leukemia and lymphoma cells [87]. SINE KPT-185, an inhibitor of exportin 1, blocks the nuclear export of FBXW7 and thereby promotes degradation of Notch1 in the nucleus of pancreatic cancer cells [88]. Genistein down-regulates expression of the microRNA miR-223, resulting in up-regulation of FBXW7 expression, inhibition of cell growth and invasion, and induction of apoptosis in pancreatic cancer cells [89].

Although most research on the role of FBXW7 in cancer has focused on its tumor suppressor function in cancer cells themselves, FBXW7 also suppresses cancer metastasis by inhibiting cancer niche formation by noncancer cells [90]. Ablation of *Fbxw7* in mouse bone marrow-derived stromal cells was thus found to promote metastatic tumor growth in a manner dependent on Notch1 accumulation and activation of *Ccl2* transcription. Increased circulating levels of the chemokine CCL2 in the FBXW7-deficient mice supported the recruitment of both monocytic myeloid-derived suppressor cells and macrophages to establish the cancer niche. Consistent with these findings in mice, low *FBXW7* expression levels in peripheral blood cells are associated with poor prognosis in human breast cancer patients [90].

### 2.4. FBXW1 (β-TrCP1) and FBXW11 (β-TrCP2)

Human β-transducin repeat containing protein (β-TrCP), which includes both β-TrCP1 (FBXW1) and β-TrCP2 (FBXW11), was originally identified as a cellular ubiquitin ligase that is bound by the HIV-1 protein Vpu and thereby directed to eliminate cellular CD4 by targeting it for proteolysis [91]. Subsequently, β-TrCP was shown to regulate multiple cellular processes by mediating the degradation of various substrates in a phosphorylation-dependent manner [5]. The WD40-repeat motif of β-TrCP recognizes a DSGX*_n_*S (where X is any amino acid) destruction motif (or variants thereof) of target proteins in which the serine residues are phosphorylated by specific kinases. Several proteins including the transcriptional coactivator β-catenin [92,93,94,95], Nuclear factor-κB (NF-κB; p105 and p100), Inhibitor of NF-κB (IκB; IκBα, IκBβ, and IκBε) [92,95,96,97,98,99,100], Cell division cycle 25 (CDC25; CDC25A and CDC25B) [101,102,103,104], Mitosis inhibitor protein kinase WEE1 [105], and Snail zinc-finger transcription factors [106,107] have been implicated as substrates of β-TrCP.

Given the diversity of its substrates, β-TrCP might be expected to play both oncogenic and tumor-suppressive roles in a context-dependent manner. Some studies have suggested that β-TrCP primarily plays an oncogenic role by mediating the degradation of tumor suppressors including IκB [92,95,96,97,98,99,100], FOXO3 [108], REST [109], and PDCD4 [110]. Indeed, β-TrCP1 is overexpressed in various types of cancer including colorectal cancer [111], pancreatic cancer [112], melanoma [113], and hepatoblastoma [114] in a manner independent of β-catenin mutational status. Tissue-specific overexpression of exogenous β-TrCP1 in mouse mammary gland and other epithelial tissues was shown to promote cell proliferation [115] and to give rise to corresponding carcinomas including mammary, ovarian, and uterine tumors. However, mice transgenic for a dominant negative form of β-TrCP1 lacking the F-box domain were found to develop intestinal adenomas as well as hepatic and urothelial tumors [116], suggestive of an oncosuppressive function of β-TrCP.

The best-characterized cancer-suppressing function of β-TrCP relates to the degradation of β-catenin, a key molecule in the canonical Wnt signaling pathway. The constitutive activation of this pathway is thought to be the initiating event in colorectal cancer, with such activation occurring mainly as a result of mutation of *APC* (adenomatous polyposis coli gene) and consequent inhibition of GSK-3β-dependent phosphorylation of the β-catenin phosphodegron. Missense mutations of the β-catenin phosphodegron (affecting residues Asp^32^, Ser^33^, Gly^34^, Ser^37^, Thr^41^ and Ser^45^) have been identified in some individuals with tumors harboring wild-type alleles of *APC* (Figure 2A), with these mutations resulting in the abnormal stabilization of β-catenin and Wnt pathway activation. Such phosphodegron mutations of β-catenin have also been detected in endometrial cancer, hepatobiliary cancer, liver cancer, and melanoma [117]. In-frame deletions spanning exon 3 of the β-catenin gene (*CTNNB1*), which encodes the phosphodegron, were recently identified in metastatic colorectal cancer, with nuclear staining of β-catenin being apparent in such tumors [118].

SETBP1 is another substrate of β-TrCP in which cancer-associated mutations are concentrated in the phosphodegron for SCF^β-TrCP^-dependent degradation (Figure 2B). SETBP1 was first identified as a protein that binds to the acute undifferentiated leukemia-associated protein SET, which inhibits the activity of protein phosphatase 2A [119], and it was subsequently found to function as a transcription factor that increases the expression of HOXA9, HOXA10 [120], and b-Myb [121] genes as well as represses that of the RUNX1 gene [122]. Somatic mutations of *SETBP1* have been identified in several myelodysplastic/myeloproliferative neoplasms, including atypical chronic myeloid leukemia [123], chronic myelomonocytic leukemia [124], juvenile myelomonocytic leukemia [125], and secondary acute myeloid leukemia [124], but not in lymphoblastic leukemia or solid tumors. Consistent with these clinical observations, transplantation experiments in mouse models revealed the biological relevance of mutated SETBP1 to myeloid leukemogenesis [124].

Although several drugs have been found to inhibit or upregulate β-TrCP, such inhibition in the clinical setting is likely to be associated with side effects, given that the substrates of β-TrCP include both tumor promoters and tumor suppressors. Indeed, treatment with the β-TrCP inhibitor erioflorin resulted in the accumulation of β-catenin, IκBα, and PDCD4 in HEK293T cells, suggesting that erioflorin stabilizes various β-TrCP targets [126]. Another β-TrCP–targeting agent, STG28, increases the abundance of β-TrCP and thereby reduces that of multiple β-TrCP substrates including β-catenin, IκBα, WEE1, CDC25A, and NF-κB (p105) [127]. However, GS143 specifically inhibits the interaction between β-TrCP and phosphorylated IκBα and thereby attenuates the ubiquitylation of IκBα, without affecting that of β-catenin [128]. Inhibitors such as GS143 that target specific substrates are more likely to have therapeutic applications.

### 2.5. FBXL5

Although iron is an essential micronutrient in cells and organisms, it can be toxic to cells if present in excess. It is therefore important that cellular iron levels be subject to strict regulation. FBXL5 is an important E3 ligase with regard to the regulation of iron metabolism [129,130]. The NH_2_-terminal region of FBXL5 contains a hemerythrin (Hr)-like domain that functions as a sensor for cellular iron. In the presence of iron, this domain forms a compact tertiary structure that is resistant to limited proteolysis and masks a degron composed of residues 77 to 81 within the Hr domain, resulting in the accumulation of FBXL5. When iron is limiting, however, the compact structure of the Hr domain is compromised and the degron becomes accessible to an as yet unidentified E3 ubiquitin ligase that mediates the polyubiquitylation of FBXL5 [131]. Various proteins including Dynactin subunit p150*^Glued^* [132], Iron-regulatory protein 1 (IRP1) [130], IRP2 [129,130], Snail [133], Single-stranded DNA-binding protein 1 (SSB1) [134], and CBP/p300-interacting-transactivator-with-an ED-rich-tail 2 (CITED2) [135] have been identified as substrates of SCF^FBXL5^.

Mice deficient in FBXL5 die in utero as a result of excessive iron accumulation [136]. This embryonic mortality was prevented by additional ablation of the FBXL5 substrate IRP2 (iron regulatory protein 2), suggesting that impaired degradation of IRP2 is primarily responsible for the death of FBXL5-deficient mice. Deletion of *Fbxl5* specifically in the hematopoietic system of mice resulted in cellular iron overload in hematopoietic stem cells and impaired the ability of these cells to repopulate bone marrow as a result of abnormal activation of both oxidative stress responses and the cell cycle [137]. Liver-specific deletion of *Fbxl5* resulted in dysregulation of both hepatic and systemic iron homeostasis, leading to the development of steatohepatitis. FBXL5 deficiency in the liver promoted chemical carcinogen-induced hepatocarcinogenesis as a result of excessive iron accumulation followed by oxidative stress, tissue damage, inflammation, and compensatory proliferation of hepatocytes [138]. Consistent with these observations in mice, down-regulation of FBXL5 expression was found to be associated with poor prognosis in patients with hepatocellular carcinoma. The abundance of *FBXL5* mRNA was also found to be reduced in the liver of patients with chronic hepatitis C virus (HCV) infection [139], and FBXL5 deficiency promotes carcinogenesis induced by the HCV core protein [138].

FBXL5 inhibits epithelial-to-mesenchymal transition (EMT) and attenuates metastasis by targeting the transcription factor Snail in gastric cancer [133,140]. It also attenuates cisplatin resistance induced by RhoGDI2 (a guanine nucleotide dissociation inhibitor for Rho-family GTPases) in gastric cancer cells [141]. In contrast to its tumor suppressor function in the liver and stomach, FBXL5 promotes colon cancer progression through regulation of the PTEN-PI3K-AKT signaling pathway [142]. FBXL5 is highly expressed in colon cancer, and its high expression has been associated with reduced overall survival and exaggerated clinicopathologic characteristics in colon cancer patients. AM404, a metabolite of acetaminophen with antibacterial activity, was found to suppress the expression of FBXL5 and to inhibit the dedifferentiation and acquisition of stem-like properties in organoids of colon cancer patients, suggesting that this agent might have potential as an anticancer drug [143]. It thus appears that FBXL5 positively or negatively regulates carcinogenesis in a tissue-dependent manner.

## 3. Tumor Promoters Regulated by Multiple F-Box Proteins

### 3.1. c-Myc

c-Myc was identified as a cellular counterpart of the v-Myc protein of avian leukemia virus. Translocation of the c-Myc gene (*MYC*) is a cause of Burkitt’s lymphoma, and the gene has long been known to play a major role in tumorigenesis. c-Myc functions as a transcription factor in association with its primary partner MAX, its interaction with which is mediated by its COOH-terminal basic helix-loop-helix/leucine zipper (bHLH/LZ) domain. Many target genes are transcriptionally activated as a result of the association of the c-Myc/MAX heterodimer with canonical E-box (CACGTG) elements within their promoter or enhancer regions. c-Myc coordinately regulates the expression of a large number of genes implicated in diverse cellular processes, including cell cycle progression, metabolism, nucleotide biosynthesis, RNA processing, translation, cell differentiation, cell senescence, and apoptosis. It is therefore not surprising that the level and activity of c-Myc are strictly regulated at the transcriptional, translational, and posttranslational levels. In particular, ubiquitylation-dependent protein degradation plays a central role in the regulation of c-Myc abundance, with the loss of such regulation leading to cancer progression.

The most frequent *MYC* mutations associated with cancer are located in the codons for Thr^58^ and Pro^59^ in the NH_2_-terminal transactivation domain, with such mutations having been shown to stabilize c-Myc by preventing its degradation by the ubiquitin-proteasome system (Figure 3A). Indeed, mice transgenic for the c-Myc(T58A) mutant show a higher penetrance and reduced latency of cancer development compared with those transgenic for the wild-type protein [144,145]. This region of c-Myc has been shown to be recognized by three F-box proteins—FBXW7, FBXL3, and FBXL14 (Figure 3A)—with SCF^FBXW7^ being the best characterized ubiquitin ligase for c-Myc degradation [68,69]. Both Thr^58^ and Pro^59^ of c-Myc are critical residues of the FBXW7 recognition motif (CPD), and phosphorylation of Thr^58^ by GSK-3β is essential for the interaction of c-Myc with FBXW7 and its subsequent degradation. Conditional deletion of *Fbxw7* in mouse thymus results in excessive accumulation of c-Myc, giving rise to marked thymic hyperplasia followed by the spontaneous development of thymic lymphoma [146]. FBXW7-mediated c-Myc degradation is also important for the maintenance of cell stemness, at least in hematopoietic stem cells [147,148,149,150], keratinocytes [151], and cancer stem cells [152,153].

FBXL3 is another F-box protein that targets the Thr^58^-phosphorylated form of c-Myc for ubiquitylation [154]. SCF^FBXL3^ was originally identified as a ubiquitin ligase for the circadian clock components CRY1 and CRY2 that functions in a clock-dependent manner [155,156,157]. CRY2, but not CRY1, was shown to cooperate with FBXL3 to affect the degradation of c-Myc in a Thr^58^ phosphorylation-dependent manner [154]. This unexpected function of CRY2 may contribute to circadian protection from tumorigenesis. FBXL14 also targets Thr^58^-phosphorylated c-Myc for ubiquitylation [158]. FBXL14 is preferentially expressed in non-stem- like glioma cells and neural progenitors, whereas it is expressed at only a low level in glioma stem cells that contribute to tumor initiation and malignant progression. Overexpression of FBXL14 induces the differentiation of and inhibits tumor formation by glioma stem cells, with these effects being reversed by expression of the c-Myc(T58A) mutant.

c-Myc has also been shown to be ubiquitylated in a manner independent of Thr^58^ phosphorylation. SCF^FBXO32^ targets c-Myc for proteasomal degradation by mediating its ubiquitylation at Lys^326^ via K48-linked ubiquitin chains [159]. FBXO32 interacts with c-Myc at its Myc-box (MB) 2, MB4, and PEST (rich in proline, glutamic acid, serine, and threonine) domains. Of note, *FBXO32* is a target gene of c-Myc, with FBXO32 and c-Myc thus contributing to a negative regulatory loop for the control of c-Myc function. FBXO28 is another F-box protein that interacts with and thereby mediates the ubiquitylation of c-Myc in a Thr^58^ phosphorylation–independent manner [160]. SCF^FBXO28^ mediates nonproteolytic ubiquitylation of c-Myc that promotes recruitment of the p300 histone acetyltransferase by c-Myc located at the promoters of its target genes. SKP2 also binds to the MB2 and bHLH/LZ domains of c-Myc associated with chromatin, which initially increases the transactivation activity of c-Myc but later results in its ubiquitylation and degradation during the G_1_-S transition of the cell cycle [161,162]. The SKP2-mediated activation of c-Myc is independent of c-Myc ubiquitylation, given that a SKP2 mutant that is unable to form an SCF complex also manifested this effect [163,164].

Although SCF^β-TrCP^ also ubiquitylates c-Myc in a manner dependent on its phosphorylation by Polo-like kinase 1 (PLK1), it forms a heterotypic K33/K48/K63 polyubiquitin chain on c-Myc instead of a monotypic K48-linked one [165]. This heterotypic polyubiquitin chain serves as a signal not for translocation to the proteasome but rather for the stabilization of c-Myc. Given that SCF^β-TrCP^ ubiquitylates the same amino acid residues of c-Myc as does SCF^FBXW7^, competition between these two F-box proteins contributes to the strict regulation of c-Myc abundance. Many other types of ubiquitin ligase including Cullin-RING ligase 3/speckle-type POZ protein (CRL3^SPOP^), Cullin-RING ligase 4/tumor necrosis factor receptor-associated ubiquitous scaffolding and signaling protein (CRL4^TRUSS^), Tripartite motif protein (TRIM32), Ring finger protein 12 (also known as RLIM), p53-induced RING H2 protein (PIRH2), HECT, UBA and WWE domain containing E3 ubiquitin protein ligase 1 (HUWE1, also known as ARF-BP1), the carboxyl terminus of Hsp70-interacting protein (CHIP), and eleven–nineteen lysine-rich leukaemia (ELL) have also been shown to mediate c-Myc ubiquitylation [166].

### 3.2. Cyclin D1

Human cyclin D1 was first discovered through screening of a cDNA expression library constructed from a glioblastoma cell line for the ability to complement the growth of conditional G_1_ cyclin-defective yeast strains [167]. D-type cyclins—including cyclins D1, D2, and D3—form active complexes with CDK4 or CDK6 that phosphorylate retinoblastoma protein (Rb) and promote G_1_-to-S phase progression [168]. They coordinate cell cycle progression with extracellular stimuli such as those related to growth factor or nutrient availability and integrin-mediated adhesion signaling [169]. Given this role of D-type cyclins, it is not surprising that their overexpression or overactivation of their cognate CDKs directly contributes to tumor growth. Indeed, the *CCND1* gene is amplified in a variety of human cancers [170].

Cyclin D1 is highly unstable, with a half-life of 10 to 30 min, and its degradation depends on the stage of the cell cycle [171,172]. The most frequently mutated codons of *CCND1* include those for Thr^286^ and Pro^287^ in the COOH-terminal region of the encoded protein (Figure 3B), with mutations affecting Thr^286^ resulting in stabilization and persistent nuclear localization of cyclin D1 [173]. Phosphorylation of Thr^286^ occurs during the G_1_-S transition and directs the CRM1-mediated export of cyclin D1 from the nucleus to the cytoplasm, where it is subjected to ubiquitylation and proteasomal degradation [171,174]. Such phosphorylation of cyclin D1 was shown to be required for ubiquitylation by four cytosolic F-box proteins: FBXO4, FBXO31, β-TrCP, and FBXW8. The interaction between FBXO4 and cyclin D1 in the cytoplasm depends mainly on the presence of αB-crystallin, which belongs to the small heat shock protein family and functions as a molecular chaperone [175]. *Fbxo4*^−/−^ or *Fbxo4*^+/−^ mice spontaneously develop multiple tumors—including lymphoma, histiocytic sarcoma, and, less frequently, mammary and hepatocellular carcinomas [176]—as well as show an increased susceptibility to the *N*-nitrosomethylbenzylamine-induced development of papilloma [177]. Loss of *Fbxo4* also increases the aggressiveness of *BRAF*^V600E^-dependent metastatic melanoma in mice [178], highlighting the importance of FBXO4 as a suppressor of tumor progression.

Whereas FBXO4 is thought to contribute to the normal cell cycle-dependent oscillation of cyclin D1, FBXO31 is a checkpoint protein induced by DNA damage that promotes cyclin D1 degradation and subsequent G_1_ arrest of the cell cycle in response to genotoxic stress [179]. The interaction of cyclin D1 with FBXO31 in vitro is independent of the phosphorylation of Thr^286^, but it requires the COOH-terminal region (residues 287−295) of cyclin D1 [180]. Biochemical analyses revealed that FBXO31 directly interacts with cyclin D1, ubiquitylates it, and targets it to the proteasome for degradation in a Thr^286^ phosphorylation-dependent manner in cells, however, suggesting that the nuclear export of cyclin D1 is important for SCF^FBXO31^-mediated ubiquitylation.

β-TrCP–mediated cyclin D1 ubiquitylation and degradation were discovered in cells treated with STG28, a derivative of troglitazone, as a result of the finding that the glitazone family of peroxisome proliferator-activated receptor γ (PPARγ) agonists induces PPARγ-independent degradation of cyclin D1 [181]. STG28 up-regulates β-TrCP expression, with the interaction of cyclin D1 and β-TrCP being promoted in STG28-treated cells. Cyclin D1 contains an unconventional recognition sequence (279-EEVDLACpT-286) for β-TrCP, with Glu^280^ serving as a phosphomimetic residue in place of the upstream phosphoserine. FBXW8 also ubiquitylates cyclin D1 in a Thr^286^ phosphorylation-dependent manner, with depletion of FBXW8 inducing marked accumulation of cyclin D1 [182]. Although SKP2 has been implicated in the regulation of cyclin D1 abundance [25], ubiquitylation of cyclin D1 by SCF^SKP2^ has not been demonstrated, suggesting that this regulation may be indirect. In addition to F-box proteins, APC/C has been shown to mediate cyclin D1 ubiquitylation [183].

Generation of *Fbxo4*^−/−^, *Fbxw8*^−/−^, and *Fbxo4*^−/−^*Fbxw8*^−/−^ mice in an investigation into which F-box proteins are most relevant to cyclin D1 degradation revealed that the stability of cyclin D1 was unchanged in cells of these animals [184]. Additional depletion of SKP2 and FBXO31 in *Fbxo4*^−/−^*Fbxw8*^−/−^ mouse embryonic fibroblasts also did not affect the half-life of cyclin D1, and interaction of cyclin D1 with either FBXO4, FBXW8, FBXO31, SKP2, or β-TrCP1 was not detected in NIH 3T3 cells. Furthermore, abrogation of the function of SCF or APC/C complexes by expression of a dominant negative mutant of CUL1 or depletion of APC2, respectively, had no effect on the half-life of cyclin D1. Together, these results suggested that the examined F-box proteins are all dispensable for cyclin D1 degradation, at least during normal progression of the cell cycle, and they thus implicated the existence of an unidentified ubiquitin ligase for cyclin D1.

### 3.3. Snail

Snail (SNAI1) was first identified in *Drosophila melanogaster* as a factor essential for mesoderm formation [185,186] and was later shown to contribute to EMT, which plays a key role in the progression and metastasis of epithelial tumors. Snail interacts with genomic DNA containing E-box sequence motifs via its COOH-terminal zinc-finger domain as well as with the histone deacetylases HDAC1 and HDAC2 via its NH_2_-terminal SNAG domain [187,188]. It acts as a transcriptional repressor—both by inducing the removal of transcriptional activators from its binding sites and by promoting histone deacetylation—at target genes such as that for E-cadherin, a marker of EMT. Expression of *SNAI1* is positively associated with tumor grade, recurrence, and metastasis as well as with poor prognosis in individuals with various tumor types [189].

Snail is highly unstable, with a half-life of 25 min in human cells [106]. Many F-box proteins (β-TrCP, FBXO31, FBXO22, FBXL14, FBXL5, FBXW7, FBXO11, and FBXO45) have been implicated in EMT on the basis of their relation to the degradation of Snail (Figure 3C). Snail is one of the best characterized targets of SCF^β-TrCP^ [106,107], with its phosphorylation at Ser^104^ and Ser^107^ promoting its localization to the cytoplasm, where its DSG phosphodegron motif (95-DpSGKGpS-100) is recognized by β-TrCP [106,190]. Phosphorylation of the Ser^104^ residue in the nucleus primes further cytosolic phosphorylation at Ser^96^ and Ser^100^ by GSK-3β, which facilitates recognition by β-TrCP. Phosphorylation at these sites is also required for FBXO31- and FBXO22-dependent ubiquitylation and degradation of Snail in gastric [191] and breast [192] cancer, respectively. Snail binds to the F-box domain of FBXO31 [191], however, is suggestive of SCF complex-independent ubiquitylation by FBXO31. Phosphorylation of Snail at Thr^203^ by LATS2 [193], at Ser^246^ by PAK1 [194], or at Ser^82^ and Ser^104^ by ERK2 [195] promotes the nuclear retention of Snail, which prolongs its half-life. Glycosylation of Snail at Ser^112^ suppresses GSK-3β-mediated phosphorylation and thereby stabilizes the protein [196]. SCF^FBXL14^ also ubiquitylates Snail in the cytosol and triggers its degradation in a manner independent of GSK-3β-dependent phosphorylation [197].

Snail is also ubiquitylated in the nucleus. Nuclear FBXL5 thus interacts with and mediates the ubiquitylation of Snail, thereby interfering with its binding to DNA [133]. Although this ubiquitylation by SCF^FBXL5^ occurs in the nucleus, Snail is subsequently exported from the nucleus and degraded in the cytosol. FBXW7 is another F-box protein that was shown to mediate the ubiquitylation of Snail in the nucleus [198,199]. Snail contains a sequence similar to the CPD (102-PPSPPSPAPS-111), although it has not been demonstrated whether this sequence is essential for the binding of Snail to FBXW7. SCF^FBXO11^ recognizes Snail via its NH_2_-terminal SNAG domain [200,201], and FBXO11 deficiency in mice results in neonatal mortality and epidermal thickening in association with the accumulation of Snail family proteins in the epidermis [201]. The atypical F-box protein FBXO45, which does not form an SCF complex but instead associates with the RING finger-type ubiquitin ligase PAM via the COOH-terminal Repeats in splA and ryanodine receptors (SPRY) domain of the latter [202], interacts with Snail via its NH_2_-terminal F-box domain [203]. SCF^FBXO45^, SCF^FBXL14^, and SCF^β-TrCP^ also ubiquitylate the EMT-related transcription factors Slug (SNAI2) and Twist1, suggesting that the corresponding F-box proteins might coordinately regulate EMT [203,204].

Although many F-box proteins contribute to the degradation of Snail, no concentration of mutations in specific regions of the protein is evident in cancer patients (Figure 3C), unlike the situation for c-Myc and cyclin D1. This difference suggests that the impaired degradation of Snail may not be critical for carcinogenesis or cancer progression, and that it may instead contribute to the acquisition of malignant traits characterized by EMT. It is thus possible that specific analysis of advanced cancer cells may reveal the regional concentration of mutations that suppress Snail degradation.

## 4. Tumor Suppressors Regulated by F-Box Proteins

### 4.1. p53

The p53 protein was initially identified in a complex with the large T antigen of simian virus 40 in transformed rodent cells [205,206], and it was first recognized as a tumor suppressor in 1989 [207]. Indeed, the p53 gene is one of the most important tumor suppressor genes, given that it is mutated in >50% of all human cancers, a mutation frequency greater than that of any other gene [208]. p53 is a transcription factor that regulates the expression of a wide spectrum of genes that contribute to various cellular functions including apoptosis, cell cycle arrest, senescence, autophagy, DNA repair, angiogenesis, and genome maintenance [209]. In the presence of low-level carcinogenic or genotoxic stimuli, p53 is persistently expressed at a low level. However, in response to cellular stresses including DNA damage, hypoxia, oncogene activation, and ribosomal stress, the degradation of p53 is inhibited, resulting in its stabilization and up-regulation of its transcriptional activity [210]. Many E3 ligases have been found to mediate the K48-linked (including Mouse double minute 2 homolog (MDM2) [211,212], PIRH2 [213], TRIM24 [214], CRL4 [215], CRL5 [216], synoviolin [217], constitutively photomorphogenic 1 (COP1) [218], Caspase 8/10-associated RING proteins (CARPs) [219], ARF-BP1 [220,221], WW domain-containing E3 ubiquitin protein ligase 1 (WWP1) [222], and CHIP [223,224]) or K63-linked (such as Tumor necrosis factor receptor-associated factor 6 (TRAF6) [225], TRIM45 [226], Fragile-site associated tumor suppressor (FATS) [227], and CHIP [228]) ubiquitylation of p53.

FBXO22 was first identified as an F-box protein that targets p53 [229] and was later shown to target methylated p53, but not acetylated p53, for degradation in senescent cells [230]. FBXO22 forms a complex with the histone demethylase KDM4 [231], and degradation of p53 dependent on the SCF^FBXO22^-KDM4A complex is essential for induction of the CDK inhibitor p16 and the senescence-associated secretory phenotype [230]. FBXO22 has been implicated as a promoter of tumorigenesis in liver cancer [232], lung cancer [233], and breast cancer [192], suggesting that ubiquitylation of p53 by SCF^FBXO22^ might underlie such a role.

SCF^FBXW7^ was recently shown to contribute to the degradation of p53 [234,235]. Given that SCF^FBXW7^-specific targeting of p53 is essential for the recovery of cell proliferation after DNA damage, inhibition of FBXW7 function might sensitize cancer cells to irradiation or chemotherapy by stabilizing p53 and supporting its induction of cell cycle arrest and apoptosis.

### 4.2. p27^KIP1^

The CDK inhibitor p27*^KIP1^* was initially identified in cells arrested by exposure to transforming growth factor–β, contact inhibition, or treatment with lovastatin [236,237,238,239]. It negatively regulates the cyclin D–CDK4 complex in G_1_ phase [240,241]. A low level of p27^KIP1^ expression, together with deregulation of CDK activity, is associated with poor prognosis in a variety of cancers, including those of the colon [242], lung [243], and stomach [244]. Down-regulation of p27*^KIP1^* in cancer is attributable to increased turnover mediated by the ubiquitin-proteasome system, rather than to inactivation or point mutation of the p27*^KIP1^* gene [27].

SCF^SKP2^ is the best characterized E3 ligase for p27*^KIP1^* degradation. p27*^KIP1^* is phosphorylated by cyclin E–CDK2 at Thr^187^ and is thereby recruited to the nucleus. SKP2 recognizes the phosphorylated form of p27*^KIP1^* and binds to its CDK-interacting site in collaboration with CKS1 in order to mediate polyubiquitylation of p27*^KIP1^* at Lys^165^ in S-G_2_ phase [245,246]. Overexpression of SKP2 has been found to correlate with reduced levels of p27*^KIP1^* and is a negative prognostic factor in several human cancers including those of the lung [247] and colon [248]. Degradation of p27*^KIP1^* in G_1_ phase is mediated by another ubiquitin ligase, Kip1 ubiquitination-promoting complex (KPC), which consists of the RING-finger catalytic subunit KPC1 and the adaptor protein KPC2 [249]. KPC ubiquitylates p27*^KIP1^* in the cytoplasm in a manner dependent on Ser^10^ phosphorylation. PIRH2 has also been identified as an E3 ligase for p27*^KIP1^* [250]. PIRH2 is expressed in late G_1_ to S phase, and it targets both nuclear and cytoplasmic p27*^KIP1^* in a phosphorylation-independent manner.

SCF^FBXL12^ was recently found to ubiquitylate p27*^KIP1^* during T cell development [251]. FBXL12 is closely related to SKP2 [4], and SCF^FBXL12^ ubiquitylates p27*^KIP1^* at Lys^165^, the same site targeted by SCF^SKP2^ [251]. Notch signaling induced transcription of the SKP2 gene, but not that of the FBXL12 gene, whereas pre-TCR (T cell receptor) signaling-induced expression of the FBXL12 gene but not that of the SKP2 gene. The absence of either SKP2 or FBXL12 in thymocytes similarly attenuated cell proliferation and differentiation in association with the accumulation of p27*^KIP1^*, and the absence of both F-box proteins showed an additive effect. These findings suggest that SKP2 and FBXL12 are both required for T cell development in an identical and additive manner as result of their targeting the same protein. Consideration of the potential of inhibition of the SKP2-p27*^KIP1^* pathway for the treatment of T cell leukemia [252] should thus take into account the role of FBXL12.

### 4.3. NRF2

NF-E2-related factor 2 (NRF2) was first identified as a homolog of the transcription factor NF-E2 and found to interact with the NF-E2 binding site [253]. It was later shown to be a master transcriptional regulator of the cellular antioxidant response. Under normal conditions, NRF2 is constitutively polyubiquitylated by the CRL3^KEAP1^ ubiquitin ligase and thereby targeted for proteasomal degradation. Exposure of cells to oxidative or electrophilic stress, however, results in the direct modification of highly reactive thiols in KEAP1 and consequent inactivation of CRL3^KEAP1^ and stabilization of NRF2 followed by its translocation to the nucleus, where it induces the expression of a battery of cytoprotective genes [254]. Carcinogenesis induced by polycyclic hydrocarbons or nitrosamine was markedly increased in Nrf2-deficient mice [255,256], suggesting that transient NRF2 activation effectively prevents chemical carcinogenesis by increasing antioxidant capabilities. In contrast, however, recent studies also described tumor-promoting roles of NRF2 activation [257,258]. Persistent NRF2 activation in cancer cells has been shown to confer therapeutic resistance against cancer agents and aggressive properties.

In addition to CRL3^KEAP1^, NRF2 is also ubiquitylated by SCF^β-TrCP^ [259,260]. NRF2 harbors a redox-insensitive degron in its Neh6 domain [261] that contains two DSG phosphodegron motifs (343-DpSGIpS-347 and 382-DpSAPGpS-387) targeted by GSK-3β. The ubiquitylation of NRF2 by SCF^β-TrCP^ is independent of CRL3^KEAP1^ activity, with the result that NRF2 can be down-regulated by activation of GSK-3β even in the presence of oxidative stress (when KEAP1 is inactivated), with such down-regulation being a potential approach to prevention of therapeutic resistance due to sustained NRF2 activation.

SCF^FBXO22^ was recently shown to mediate the ubiquitylation and consequent degradation of BTB domain and CNC homolog 1 (BACH1), a downstream effector of NRF2 that contributes to control of the transcription of antioxidant genes by NRF2 in a heme-dependent manner [262]. The accumulation of NRF2 in lung cancer promotes the stabilization of BACH1, and the loss of KEAP1 or FBXO22 induces metastasis in a BACH1-dependent manner. Two other E3 ligases, SCF^FBXL17^ and HOIL1, also ubiquitylate BACH1 [263,264] and might therefore act cooperatively with SCF^FBXO22^ to modulate the NRF2 pathway.

## 5. Conclusions

Various F-box proteins have tumor-suppressive or oncogenic functions and have therefore been proposed as potential anticancer targets, with several inhibitors of these proteins having been developed [265]. However, none of these agents have yet entered clinical studies, with the field still being at an early stage. Certain ubiquitin ligases are individually responsible for the integrative degradation of multiple substrates that contribute to a common biological process, with the result that a small number of ubiquitin ligases can degrade a wide variety of substrates. Inhibition of a given F-box protein might therefore have unintended consequences as a result of effects on multiple substrates. In addition, the stability of some individual proteins is regulated by multiple ubiquitin ligases. This situation may reflect fine-grained control of protein function in response to changes in the intracellular environment, with the potential incorporation of multitiered backup mechanisms and operation in a cell- or tissue-dependent manner. Drugs that inhibit one specific F-box protein might therefore be insufficient to halt the development or progression of cancer. It is important, however, to rule out the possibility that experimental findings in this regard are not artifacts due to inadequate analysis. One effective approach to determining which ubiquitin ligase–substrate relations are most relevant to cancer progression is to integrate information on genetic alterations in cancer patients and the results of biochemical analysis, as illustrated by the examples provided in this review. The accumulation of such findings is expected to advance the development of new cancer therapeutics.

## Figures and Tables

**Figure 1 cancers-12-01249-f001:**
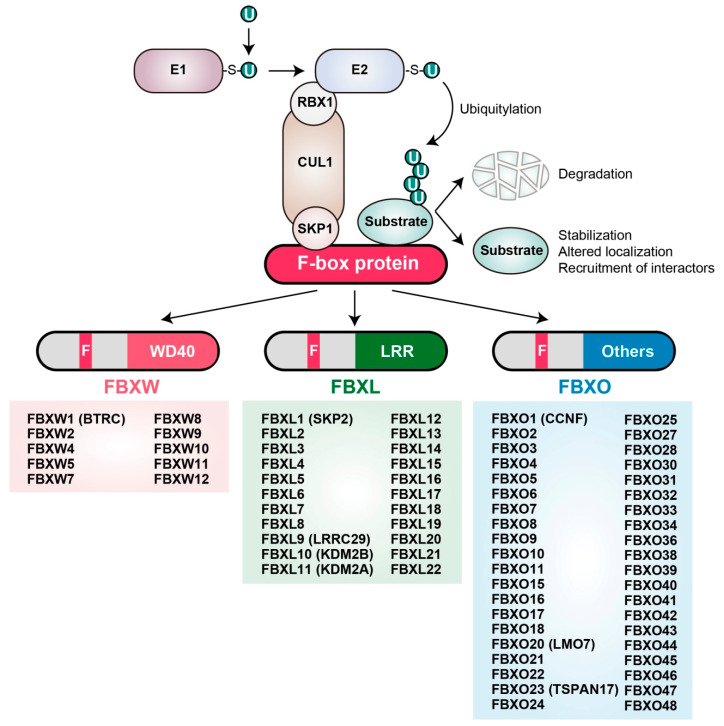
The S-phase kinase-associated protein 1 (SKP1)-Cullin1 (CUL1)-F-box protein (SCF) complex and its ubiquitylation of target proteins. The SCF complex functions together with E1 and E2 enzymes to mediate the ubiquitylation of target proteins. Each F-box protein binds to SKP1 via its F-box domain and to a substrate via its COOH-terminal substrate interaction domain, thereby presenting the target protein for ubiquitylation. The SCF complex-mediated formation of a polyubiquitin chain on a substrate in most instances serves as a signal for proteasome-mediated degradation, although in some cases it instead leads to protein stabilization, a change in localization, or recruitment of other binding proteins (see Section 2.2 and Section 3.1). F-box proteins fall into three major classes based on the type of substrate interaction domain: those that contain WD40 repeats (FBXW), leucine-rich repeats (FBXL), or other protein interaction domains (FBXO). Human F-box proteins in each class are shown. U, ubiquitin.

**Figure 2 cancers-12-01249-f002:**
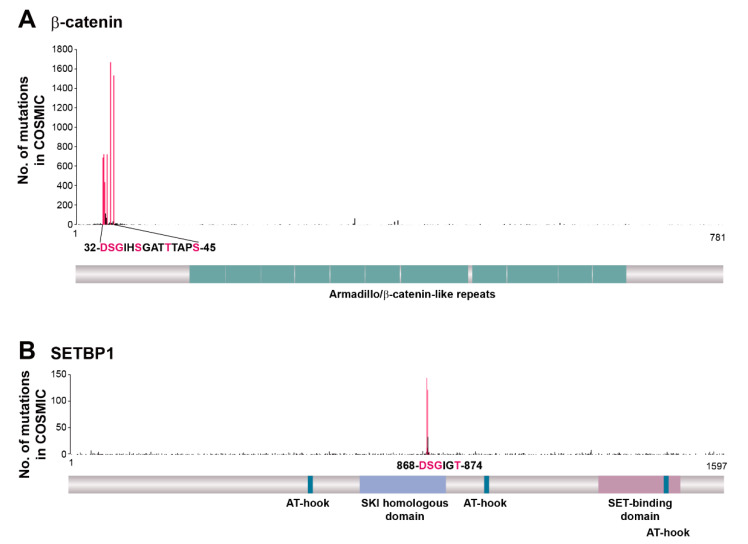
Distribution along the encoded protein sequences of *CTNNB1* (β-catenin gene) and *SETBP1* mutations identified in the genomes of cancer patients. The numbers of *CTNNB1* (**A**) and *SETBP1* (**B**) genetic alterations were obtained from the COSMIC database (version 90). Mutations corresponding to the consensus sequence of the phosphodegron for β-TrCP are shown in pink.

**Figure 3 cancers-12-01249-f003:**
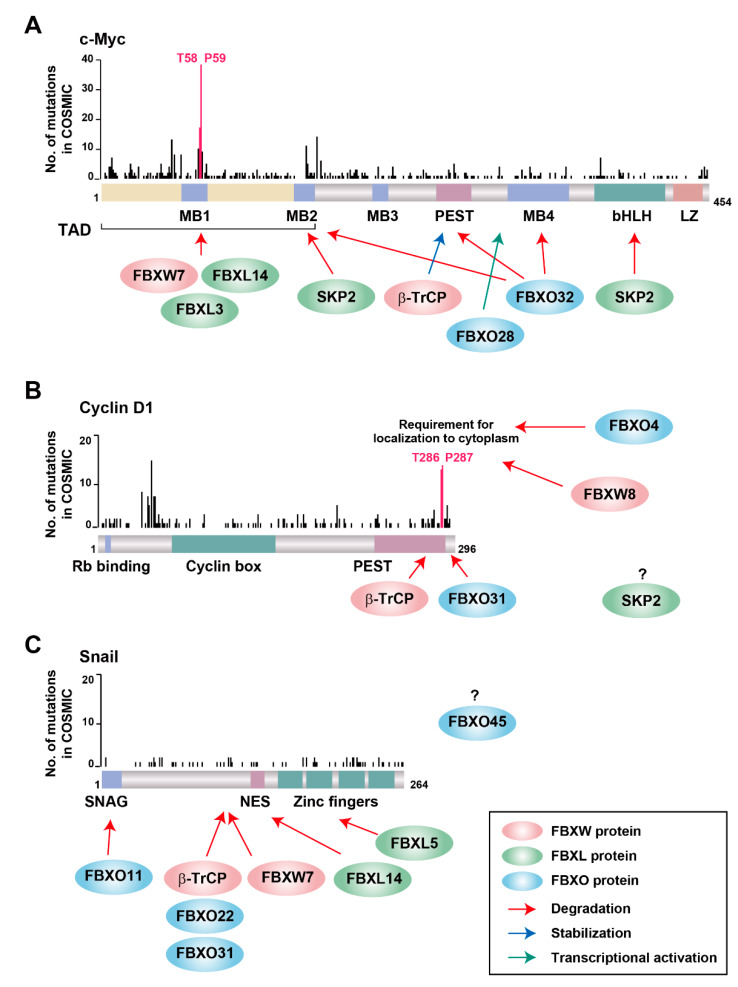
Distribution along the encoded protein sequences of *MYC*, *CCND1*, and *SNAI1* mutations identified in the genomes of cancer patients. The numbers of *MYC* (**A**), *CCND1* (**B**), and *SNAI1* (**C**) genetic alterations were obtained from the COSMIC database (version 90). Mutations corresponding to critical amino acids for the half-life of c-Myc (**A**) or cyclin D1 (**B**) are shown in pink. Arrows show the recognition sites for F-box proteins and their cofactors.

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
