# Peer review of "F-Box Proteins and Cancer"

_cancers, 2020, doi:10.3390/cancers12051249_

Round 1

Reviewer 1 Report

In this manuscript, the association between F-box proteins in SCF E3 ligases and cancer was reviewed. The SCF complex is a huge family, which is a major component in ubiquitin pathway, functioning in both degradation and non-degradation manners. The representative members were discussed in this draft. SCFs were involved in all aspects of cancer progression, of which the key tumor promoters targeted by SCFs were discussed.

However, it is a partially reviewed paper when it is about cancer.

  1. There are also tumor suppressors (e.g. p53) which are equally important in cancer progression. It would be interesting to summarize how they were regulated by F-proteins.
  2. Another flaw for a review relating to cancer is it lacks information of therapeutics relating to F-box proteins. I would like to see the top picks of targeted F-box proteins in cancer therapy.

One confusing information in Line 31: There are RING E3s and HECT E3s. They transfer ubiquitin in different mechanisms.

Author Response

Response to Reviewer #1

We thank the reviewer for the careful evaluation of our manuscript and for suggestions that we feel have helped us to improve our manuscript.

Our specific responses to the points raised are as follows:

  1. There are also tumor suppressors (e.g. p53) which are equally important in cancer progression. It would be interesting to summarize how they were regulated by F-proteins.

[Response] We agree with the reviewer that excessive degradation of tumor suppressors is also important for cancer progression. We have already mentioned that SCFSKP2 plays an important role in the degradation of CDK inhibitor p27KIP1 [ref. 1-5], p57KIP2 [ref. 6], and p21CIP [ref. 7,8], all of which are considered to be tumor suppressors. Several types of mutations and amplifications of SKP2 were found in different types of cancers—including cervical, endometrial, adrenocortical, ovarian, breast and non-small cell lung cancers [ref. 9], and the expression level of SKP2 is negatively associated with patient survival in various types of cancer—including prostate cancer [ref. 10], breast cancer [ref. 11,12], and lung cancer [ref. 13]—suggesting that SKP2 is considered to be an oncogene. We have clarified these points in the revised manuscript (page 4, lines 121-125).

              Another tumor suppressor pRb was shown to be ubiquitylated by multiple E3 ligases such as MDM2 [ref. 14], NRBE3 [ref. 15], and hUTP14a [ref. 16]. PTEN was also reported to be ubiquitylated by many E3 ligases including CRL4DCAF13 [ref. 17], XIAP [ref. 18], WWP2 [ref. 19], NEDD4-1 [ref. 20], and CHIP [ref. 21], none of which are SCF-type ubiquitin ligases. Given that we intend to focus on specific substrates that are regulated by multiple F-box proteins, we excluded these cases from this review.

              Recently, SCFFBXW7 was shown to contribute to the degradation of tumor suppressor p53 [ref. 22,23], in addition to many other ubiquitin ligases including MDM2 [ref. 24,25], PIRH2 [ref. 26], TRIM24 [ref. 27], CRL4DDB1 [ref. 28], CRL5 [ref. 29], synoviolin [ref. 30], COP1 [ref. 31], CARP1/2 [ref. 32], ARF-BP1 [ref. 33,34], WWP1 [ref. 35], and CHIP [ref. 36,37]. Given that SCFFBXW7-specific targeting of p53 is essential for the recovery of cell proliferation after DNA damage [ref. 36,37], the inhibition of FBXW7 function might sensitize cancer cells to irradiation or chemotherapy by stabilizing p53 to induce cell-cycle arrest and apoptosis. We have now addressed these points in the revised manuscript (page 5, lines 169-173).

  1. Another flaw for a review relating to cancer is it lacks information of therapeutics relating to F-box proteins. I would like to see the top picks of targeted F-box proteins in cancer therapy.

[Response] It has become clear that a number of F-box proteins have tumor suppressive or oncogenic functions. For this reason, many F-box proteins have been proposed as potential cancer therapeutic targets and several inhibitors of some F-box proteins have been developed [ref. 38], however, none of them have entered clinical studies to date, leaving a major challenge for cancer therapies targeting F-box proteins. Certain ubiquitin ligases are individually responsible for the integrative degradation of multiple substrates that contribute to a common biological process, with the result that a small number of ubiquitin ligases can degrade a wide variety of substrates. Inhibition of one particular F-box protein might therefore result in unintended consequences by affecting multiple substrates. In addition, the stability of some proteins is regulated by multiple ubiquitin ligases. This situation may reflect fine-grained control of protein function in response to changes in the intracellular environment, with the potential incorporation of multitiered backup mechanisms and operation in a cell- or tissue-dependent manner. Drugs that simply inhibit one specific F-box protein thus have the potential to be insufficient to control the development or progression of cancer. It is important, however, to exclude the possibility that experimental findings in this regard are not artifacts due to inadequate analysis. One effective approach to determining which ubiquitin ligase–substrate relations are most relevant to cancer progression is to integrate information on genetic alterations in cancer patients and the results of biochemical analysis, as illustrated by the examples provided in this review. The accumulation of such findings is expected to advance the development of new cancer therapeutics. In response to this comment, we have now added a statement regarding F-box protein-targeted therapies in the Conclusions section of the revised manuscript (page 12, lines 467-479).

  1. One confusing information in Line 31: There are RING E3s and HECT E3s. They transfer ubiquitin in different mechanisms.

[Response] As pointed out by the reviewer, RING E3 and HECT E3 transfer ubiquitin to the substrate by different mechanisms. HECT-type E3s catalyze ubiquitylation by first forming an E3-ubiquitin thioester intermediate. RING-finger, U-box, and PHD-type E3s do not appear to form such an intermediate. We have now clarified these points in the Introduction section of the revised manuscript (page 1, lines 32-34).

Reviewer 2 Report

F-Box Proteins and Cancer by Yuminoto et al is a review article that summarizes the role of selected F-box proteins which function either as tumor suppressor or promotors.  The authors then select 3 proto-oncoproteins (Snail, c-Myc, and Cyclin D1) and describe how they are regulated by their associated F-box proteins.   The manuscript is well written and provides a good overview of the F-box proteins and how they function in cancer using limited but specific examples.

Minor points.

More references in the intro.  References should be given for the sentence that ends on line 35, 39, 42 and 43.

Figure 1.  On the right hand side there is a representation of a ubiquitinated substrate being degraded or being stabilized, having an altered localization of recruiting interacting partners.  What does this refer to as it is not described in the legend or text.  The article focuses on degradation but maybe the authors are alluding to the fact that degradation is not always the outcome, as they describe for c-Myc in lines 337-340.   This should be clarified.

Figure 3.  The arrows show X-box and co-factor recognition sites for Snail, c-Myc and Cyclin D1.  The proteins are circled in green, red or blue but the significance of this is not described and is not clear.

Author Response

Response to Reviewer #2

We thank the reviewer for the careful evaluation of our manuscript and for the statement that “The manuscript is well written and provides a good overview of the F-box proteins and how they function in cancer using limited but specific examples.” We also thank the reviewer for suggestions that we feel have helped us to improve our manuscript.

Our specific responses to the points raised are as follows:

  1. More references in the intro. References should be given for the sentence that ends on line 35, 39, 42 and 43.

[Response] As suggested by the reviewer, references have been added in the revised manuscript.

  1. Figure 1. On the right hand side there is a representation of a ubiquitinated substrate being degraded or being stabilized, having an altered localization of recruiting interacting partners. What does this refer to as it is not described in the legend or text. The article focuses on degradation but maybe the authors are alluding to the fact that degradation is not always the outcome, as they describe for c-Myc in lines 337-340. This should be clarified.

[Response] We apologize for the insufficient explanation. As pointed out by the reviewer, formation of polyubiquitin chains on substrates mediated by SCF complexes serves as a signal for proteasome-mediated degradation in most cases, whereas ubiquitylation of certain substrates leads to protein stabilization, changes in localization, or recruitment of other binding proteins rather than protein degradation (see sections 2.2. FBXL1 (SKP2) and 3.1. c-Myc). We have now clarified these points in the Figure Legends of the revised manuscript (page 2, lines 54-58).

  1. Figure 3. The arrows show X-box and co-factor recognition sites for Snail, c-Myc and Cyclin D1. The proteins are circled in green, red or blue but the significance of this is not described and is not clear.

[Response] We apologize for the insufficient explanation. The proteins circled in red, green, and blue in Figure 3 represent the members of FBXW, FBXL, and FBXO subfamilies, respectively. We have now modified Figure 3 by adding an illustration to explain this in the revised manuscript (page 9, line 330).

Reviewer 3 Report

F-Box proteins and Cancer

This is a very thorough and well written from the Nakayama lab review on the various roles played by F-box proteins in cancer development. The authors first highlight representative and well characterized F-box proteins that have strong links with cancer biology. They summarize abundant and sometimes conflicting findings on the various substrates of F-box proteins that are relevant to cancer. The importance of certain substrates and their ubiquitin-mediated regulation during oncogenesis is highlighted by the abundance of mutations found in cancer samples that target regions important for their stability. This is well illustrated by a few figures in the manuscript. A final section concerning key tumor promoters that are jointly regulated by multiple SCF E3 ligases shows the intricate ways through which these key factors are regulated by the ubiquitin-proteasome pathway. All in all, this is a timely review that will greatly interest the readers of Cancers.

Typos and grammar:

Line 113. as manifested by a reduced growth rate…

Line 253. in utero

Line 374. in vitro

Author Response

Response to Reviewer #3

We thank the reviewer for the careful evaluation of our manuscript and for the statement that “This is a very thorough and well written from the Nakayama lab review on the various roles played by F-box proteins in cancer development.” and that “All in all, this is a timely review that will greatly interest the readers of Cancers.” We also thank the reviewer for suggestions that we feel have helped us to improve our manuscript.

Our specific responses to the points raised are as follows:

  1. Typos and grammar:

Line 113. as manifested by a reduced growth rate…

Line 253. in utero

Line 374. in vitro

[Response] As suggested by the reviewer, typos and grammar have been corrected in the revised manuscript.

Round 2

Reviewer 1 Report

Thanks for addressing my concerns in the first review report. The HECT E3 ligases were explained well. The other explanation from authors make scene, but however, the revision doesn’t improve much. In my opinion, it needs major rewriting to make it worth reading.

Here are my reasons:

  1. Please look back to subtitles of the paper: 1. Introduction; 2. Representative F-Box Proteins Related to Cancer; 3. Tumor Promoters Regulated by Multiple F-Box Proteins; 4. Conclusions. It is so obvious that it will be more completed if “tumor suppressors regulated by F-box proteins” added independently.
  2. F-box proteins as therapeutic targets: All the cancer studies should be ultimately translational. it is true that targeting E3 could trigger unintended consequences as it is a complicated network. That is why it is critical to review the challenges encountered in therapeutic study. If there are questions, it is the value of a review paper to bring them up. I have no problem of putting this discussion in final conclusion part, but specific examples should be discussed from original studies. Citing another review paper is not acceptable.

Author Response

Response to Reviewer #1

We thank the reviewer for the careful evaluation of our manuscript and for suggestions that we feel have helped us to improve it.

Our specific responses to the points raised are as follows:

Please look back to subtitles of the paper: 1. Introduction; 2. Representative F-Box Proteins Related to Cancer; 3. Tumor Promoters Regulated by Multiple F-Box Proteins; 4. Conclusions. It is so obvious that it will be more completed if “tumor suppressors regulated by F-box proteins” added independently.

[Response] As suggested by the reviewer, we have added a new section 4 titled “Tumor Suppressors Regulated by F-Box Proteins,” in which we describe regulation of the tumor suppressor proteins p53, p27KIP1, and NRF2 by F-box proteins.

  1. F-box proteins as therapeutic targets: All the cancer studies should be ultimately translational. it is true that targeting E3 could trigger unintended consequences as it is a complicated network. That is why it is critical to review the challenges encountered in therapeutic study. If there are questions, it is the value of a review paper to bring them up. I have no problem of putting this discussion in final conclusion part, but specific examples should be discussed from original studies. Citing another review paper is not acceptable.

[Response] Inhibitors have been developed for four of the five F-box proteins that we describe in detail, the exception being cyclin F. As suggested, we now describe these inhibitors of SKP2, FBXL5, FBXW7, and β-TrCP and their clinical potential in the corresponding sections of the revised manuscript.

Round 3

Reviewer 1 Report

Thanks for addressing my previous concerns. Now it reads more logically sound and provides more adequate information to a broader range of readers.